# LTP Allergy Follow-Up Study: Development of Allergy to New Plant Foods 10 Years Later

**DOI:** 10.3390/nu13072165

**Published:** 2021-06-24

**Authors:** Diana Betancor, Alicia Gomez-Lopez, Carlos Villalobos-Vilda, Emilio Nuñez-Borque, Sergio Fernández-Bravo, Manuel De las Heras Gozalo, Carlos Pastor-Vargas, Vanesa Esteban, Javier Cuesta-Herranz

**Affiliations:** 1Department of Allergy, Instituto de Investigación Sanitaria Hospital Universitario Fundación Jiménez Díaz (IIS-FJD, UAM), 28040 Madrid, Spain; diana13_b@hotmail.com (D.B.); alicia.gomezl@quironsalud.es (A.G.-L.); c.villalobosvilda@gmail.com (C.V.-V.); mheras@fjd.es (M.D.l.H.G.); j.cuestaherranz@gmail.com (J.C.-H.); 2Department of Immunology, Instituto de Investigación Sanitaria Hospital Universitario Fundación Jiménez Díaz (IIS-FJD, UAM), 28040 Madrid, Spain; enbli7@hotmail.com (E.N.-B.); sergio.fernandezb@quironsalud.es (S.F.-B.); vesteban@fjd.es (V.E.); 3RETIC ARADyAL, Instituto de Salud Carlos III, 28029 Madrid, Spain; 4Department of Biochemistry and Molecular Biology, Universidad Complutense de Madrid, 28040 Madrid, Spain; 5Faculty of Medicine and Biomedicine, Alfonso X El Sabio University, Villanueva de la Cañada, 28691 Madrid, Spain

**Keywords:** nsLTP, plant-food allergy, Pru p 3, peach, nut, *Rosaceae* fruit, ISAC

## Abstract

Introduction: Allergy to nonspecific lipid transfer protein (nsLTP) is the main cause of plant-food allergy in Spain. nsLTPs are widely distributed in the plant kingdom and have high cross-reactivity but extremely variable clinical expression. Little is known about the natural evolution of this allergy, which complicates management. The objective of this study was to assess the development of allergy to new plant foods in nsLTP-sensitized patients 10 years after diagnosis. Methods: One hundred fifty-one patients showing specific IgE to nsLTP determined by ISAC (Thermofisher) were included. After clinical workup (i.e., anamnesis, skin test, and challenge when needed), these patients were divided into two groups: 113 patients allergic to one or more plant food (74.5%) and 38 patients not allergic to any plant food (25.1%). Ten years later, a telephone interview was conducted to check whether patients had developed additional allergic reactions to plant foods. Results: Ten years after diagnosis, 35 of the 113 (31%) plant-food-allergic patients sensitized to nsLTP reported reactions to new, previously tolerated plant foods, mainly *Rosaceae*/*Prunoideae* fruits and nuts followed by vegetables, *Rosacea*/*Pomoideae* fruits, legumes, and cereals. Five out of 38 (13.2%) patients previously sensitized to nsLTP but without allergy to any plant food had experienced allergic reactions to some plant food: two to *Rosaceae*/*Prunoideae* fruits, two to *Rosaceae*/*Prunoideae* fruit and nuts, and one to legumes. Conclusion: Patients sensitized to nsLTP developed allergic reactions to other plant foods, mainly *Rosaceae-Prunoideae* fruits and nuts. This was more frequent among plant-food-allergic patients than among those who had never had plant-food allergy.

## 1. Introduction

Food allergy affects around 0.3% to 5.6% of the population, showing substantial geographical variation in prevalence and in terms of the culprit food [1]. Allergy to plant foods is the most common food allergy among older children and adults [1].

Nonspecific lipid transfer proteins (nsLTPs) are small, highly stable and conserved molecules involved in the plant defense against fungi and bacteria [2,3]. nsLTPs are found in high concentrations in the epidermal tissues of fruits and are the main allergens of fruits of the Rosaceae family. In addition, allergenic nsLTPs have been found in nuts, seeds, vegetables, pollen, and latex from *Hevea brasiliensis* [4]. Allergy to nsLTP involves several taxonomically unrelated plant-derived foods and heterogeneous sensitization profiles and can trigger severe systemic reactions. It has been reported to be responsible for a large number of plant-food-induced anaphylactic reactions in southern Europe [5,6,7,8].

Fruits of the Rosaceae family are the most frequently involved foods in allergic reactions among nsLTP-allergic patients [9]. Allergy to nsLTP occurs predominantly in the Mediterranean area (Spain, Italy, etc.) [5,6], although it has also been reported in other areas such as Australia [10] and China [11]; in contrast, nsLTP allergy is a rare finding in northern and central Europe [7,12] and the USA [5].

Patients with allergy to nsLTP exhibit considerable clinical heterogeneity, as some react to only one food (often peach), while others may experience symptoms to multiple nsLTPs from allergenic sources that are not taxonomically related and do not follow a defined pattern [13]. The extreme variability of nsLTP allergy in terms of the culprit plant food and the clinical expression of the allergy is still unexplained. Strict plant-food avoidance diets are sometimes recommended due to the unknown clinical course, though these measures have a significant negative impact on patients’ quality of life and nutrition. Little is known about the natural evolution of this syndrome.

The management of patients allergic to nsLTP is complex and poses a major challenge for both allergists and patients. The problem lies in the fact that LTP is a panallergen, meaning that it is a ubiquitous protein that is widely distributed in plant foods and has wide cross-reactivity and a highly variable clinical expression, sometimes eliciting life-threatening reactions. Further complicating this situation is the possibility that patients sensitized to homologous nsLTPs of other plant foods can progress over time from mere sensitization (without clinical expression) to severe or even fatal allergic reactions, which has clear implications for the dietary recommendations given to nsLTP-allergic patients.

On the other hand, the LEAP study revealed that early food introduction can prevent the onset of allergy [14], the STOP study showed that induction of tolerance can halt allergy [15], and Pru p 3 SLIT induces an improvement not only in peach allergy but also acts upon other relevant food allergens causing severe reactions, such as peanut or tree nuts [16,17,18]. These facts could also have important implications for dietary recommendations for LTP-allergic patients. In this respect, intake of plant foods containing cross-reactive proteins that the patient tolerates and to which he/she is sensitized might improve LTP allergy in the future.

The management of such patient heterogeneity continues to challenge the expertise of allergists despite the study by Asero et al. [19] and the recommendations given by the EAACI Task Force on nsLTP Allergy Across Europe [4].

The aim of this study was to assess the development of allergy to new plant foods in nsLTP-sensitized patients over 10 years. The results reinforce key points that inform decision-making related to the management of this heterogenous and complex type of allergy.

## 2. Materials and Methods

### 2.1. Study Design

One hundred fifty-one out of 164 patients sensitized to nsLTP as determined by ImmunoCAP™ ISAC (Thermo Fisher Scientific, Uppsala, Sweden) performed during 2009–2011 in the allergy department of Fundación Jiménez Díaz (Madrid, Spain) were included in the study. Thirteen patients (7.9%) were excluded because they did not respond to the follow-up phone call, refused to answer, or did not give consent to participate in the study. After a clinical study (2009–2011) in real-life conditions (i.e., anamnesis, skin test or specific IgE and challenge test when needed), patients were divided into 2 groups: 113 patients allergic to plant food (74.8%) and 38 non-food-allergic patients (25.1%). Once a patient was diagnosed with an allergy to a plant food, they were advised to avoid the food in question and continue eating those they tolerated. Ten years later, in 2020–2021, a telephone interview was conducted to determine whether the patients had developed new allergic reactions to previously tolerated plant foods (Figure 1).

### 2.2. Specific IgE to LTP

All patients showed specific IgE to at least one nsLTP (Pru p 3, Cor a 8, Art v 3 before 2011 and Pru p 3, Cor a 8, Art v 3, Ara h 9, Jug r 3, Ole e 7, Pla a 3 after 2011) measured by ImmunoCAP™ ISAC following manufacturer recommendations. Results were expressed in ISU (ISAC standardized units).

### 2.3. Study Variables

On the one hand, data were collected at the time of diagnosis, including demographic and clinical characteristics of the patients; sensitization to common allergens (defined as at least 1 positive skin prick test or serum-specific IgE to common allergens); associated rhinitis or asthma; specific IgE to different nsLTPs, profilins, and PR-10 proteins as determined by ImmunoCAP™ ISAC microarray; and data related to plant-food allergy such as the plant food eliciting allergy and symptoms of the reactions, which were categorized into local symptoms, systemic symptoms, and anaphylaxis (two or more organs involved).

On the other hand, after the telephone interview, data collected at the time of diagnosis were re-evaluated to distinguish those patients who had developed allergy to nsLTP-related foods during follow-up so as to search for characteristics that could predict progression to allergy in nsLTP syndrome.

## 3. Statistical Analysis

Statistical analysis was performed with SPSS (SPSS Inc., Chicago, IL, USA). Qualitative variables were expressed as percentages and confidence intervals were calculated at 95%. For quantitative variables, means and standard deviation (SD) were calculated, and for specific IgE results, medians and 25th (Q1) and 75th (Q3) percentiles were given. A χ^2^ test was used to compare frequencies. Values were considered significant at a *p*-value of less than 0.05.

## 4. Results

### 4.1. Patient Characteristics

One hundred fifty-one patients sensitized to nsLTP were selected and analyzed for this study. Thirty-eight patients were asymptomatic upon nsLTP-related food exposure and 113 patients were allergic to nsLTP-related plant foods. Characteristics of the patients are shown in Table 1.

#### 4.1.1. Plant-Food-Allergy Group (Baseline Data)

One hundred thirteen patients sensitized to nsLTP were allergic to plant food before the start of the follow-up period. Characteristics of the patients are shown in Table 1.

The frequency of sensitization to different nsLTPs (ISAC) at the beginning of the study was as follows: 87.6% to Pru p 3 (*n* = 99 out 113 patients tested), with a median positive test value of 3.3 ISU (1.15–5.5 Q1–Q3); 80.6% to Pla a 3 (*n* = 29/36), median 0.8 ISU (0.6–2.2); 75.9% to Jug r 3 (*n* = 23/29), median 1 ISU (0.55–1.75); 64.3% to Art v 3 (*n* = 72/112), median 1.6 ISU (0.6–3.2); 53.1%to Cor a 8 in (*n* = 59/111), median 1.3 ISU (0.7–3.13); 59.3% to Ara h 9 (*n =* 16/27), 0.9 ISU (0.5–1.6); and 30.8% to Ole e 7 (*n* = 8/26), median 1.3 ISU (0.4–2.3). These results are shown in Figure 2.

Foods eliciting allergy in this patient group are listed in Table 2. Peach and nut were the most frequently involved plant foods (70 and 54 patients, respectively) followed by apple (37 patients) and hazelnut and peanut (36 patients in both). Cofactors were associated in 11 patients (9.7%), 4 of whom had anaphylaxis.

Eighty-five patients (75.2%) developed systemic symptoms, 23 of whom (20%) experienced an anaphylactic reaction. The plant foods responsible for the anaphylactic reactions were as follows: nuts (39.1%), *Rosacea*/*Prunoideae* fruits (21.7%), *Rosacea*/*Pomoideae* fruits (17.4%), lettuce (13.0%), and legumes (8.7%). Sensitization to profilin in this anaphylaxis subgroup was 26% (6 patients) and 30.4% were sensitized to PR-10 (7 patients). The rate of anaphylaxis in profilin-sensitized patients was 7.8%, and 22% of profilin-negative patients presented anaphylaxis.

#### 4.1.2. Non-Food-Allergy Group (Baseline Data)

At the start of the study, 38 out of 151 patients sensitized to any nsLTP had not experienced any plant-food allergy. These patients made up the group of non-plant-food-allergic patients. Characteristics of the patients are shown in Table 1.

The most common allergens identified through specific IgE (ISAC) were the following: Pru p 3 in 70.3% (26/37) of patients, median 1.85 ISU (0.8–3.4); Art v 3 in 40.5% (15/37), median value 0.8 ISU (0.6–1.45); Cor a 8 in 37.84% (14/37), 1.05 ISU (0.6–1.4). These results are shown in Figure 2.

There was no statistically significant difference in specific IgE to different nsLTPS between the 2 groups (nsLTPS-allergy group and the non-food-allergy group). However, there was a statistically significant difference between the percentage of positive patients between the 2 groups to Pru p 3 (*p* = 0.012) and Art v 3 (*p* = 0.013), but not to Ara h 9, Cor a 8, Jug r 3, Ole e 7, and Pla a 3. Comparisons are shown in Figure 2.

### 4.2. Characteristics of Patients Not Sensitized to Pru p 3

Twenty-five of 156 patients had negative specific IgE to Pru p 3: 14/113 patients (12.4%) from the plant-food-allergy group and 11/38 patients (28.9%) from the group without food allergy.

Focusing on the plant-food-allergy group, 5 out of 14 (20%) patients had a systemic reaction, one of which (4%) was an anaphylactic reaction. Despite the negative value for Pru p 3, 9 patients had allergy-related symptoms to peach. Sensitization to nsLTP among Pru p 3-negative patients was as follows: 7 patients (50%) monosensitized to Art v 3, 2 patients to Pla a 3, 1 patient to Cor a 8, and 1 patient to Ara h 9. The other 3 patients were polysensitized with Art v 3 involved in all cases: 1 patient to Cor a 8 and Art v 3, 1 patient to Jug r 3 and Art v 3, and 1 patient to Ara h 9, Jug r 3, Art v 3, and Pla a 3. These results are shown in Table 3.

Nine patients non-sensitized to Pru p 3 were positive for other panallergens: 6 to profilin and 3 to the PR-10 protein family.

In the group without food allergy, 5 patients (55.4%) were monosensitized to Art v 3, 3 patients to Cor a 8, and 1 patient to Jug r 3. The other patient was sensitized to both Cor a 8 and Art v 3.

Three patients were sensitized to other panallergens: 2 patients to allergens belonging to the PR-10 protein family and 1 patient to profilin.

### 4.3. Follow-Up Study: Allergy to New Plant Foods over the Years

Forty out of 151 patients sensitized to nsLTP (26.5%; 95% CI 20–34%) developed symptoms of allergy to new (previously tolerated) plant foods during the follow-up period. Patients in this group had a mean age of 31.4 years (range 2 to 62 years) with a higher prevalence of female patients (60%). In addition, 95% of patients had a history of atopy and 90.2% had current atopy.

The frequency of sensitization to different nsLTPs (ISAC) at the beginning of the study was as follows: 85% of 40 patients were sensitized to Pru p 3, with a median value of 2.4 ISU (1.04–4.7 Q1–Q3); 50% of 10 patients to Ara h 9, median 0.9 ISU (0.4–1); 52.5% of 40 patients to Cor a 8, median 1.2 ISU (0.6–0.8); 72.7% of 11 patients to Jug r 3, median 1 ISU (0.9–2.8); 55% of 40 patients to Art v 3, median 1.2 ISU (0.6–2.1); 77% of 13 patients to Pla a 3, mv 1.56 ISU (0.6–2.3); and 11.1% to ole e 7 (one patient). There was no statistical difference in the specific IgE rate to different LTPs between the group of patients that developed allergy to new plant foods or not (Figure 2). In addition to nsLTP sensitization, 8 patients (20%) that developed allergy to new plant foods were also sensitized to PR-10 and 9 patients (22.5%) to profilin.

#### 4.3.1. Plant-Food-Allergy Group: Allergy to New Foods

Thirty-five (31%; 95% CI 23–40%) of the 113 patients from the plant-food-allergy group developed allergy to new plant foods: 16 patients to *Rosaceae* fruits (13 to *Rosaceae*/*Prunoideae* fruits and 3 to *Rosaceae*/*Pomoideae* fruits), 16 to nuts (5 patients shared *Rosaceae* fruits and nuts), 4 patients to vegetables, 2 to cereals, 1 to legumes, and 1 to seeds. The allergy symptoms in these patients were local reactions in 37.1% and systemic reactions in 62.9%; 8.6% (of the total) were anaphylactic reactions. All new plant foods that elicited allergic reactions during the follow-up period are shown in Table 4.

Patients from this group had a mean age of 26.9 years (range 2 to 61 years) and were predominantly female (60%). Sensitization to common allergens was present in 91.43% of the patients; 85.7% of the patients had associated rhinitis while 62.8% presented asthma. Sensitization to profilin was 31.4% and 17.1% were sensitized to PR-10. Nineteen patients received grass pollen immunotherapy and none of them to birch pollen.

#### 4.3.2. Non-Food-Allergy Group: Allergy to New Foods

Five out of 38 patients (13.2%; 95% CI 6–27%) from the non-food-allergy group, which comprised patients who had never experienced allergic reactions to any plant foods, developed allergy to new plant foods. The plant foods eliciting allergy in this subgroup were as follows: *Rosacea* fruits in 2 patients, nuts in 2 patients, and legumes in 1 patient. Two patients from this group developed allergy to both *Rosacea* fruits and nuts. The allergy symptoms in these patients were local in 60% and systemic in 40%. None experienced anaphylactic reactions.

Patients from this group ranged in age from 18 to 50 years (mean age 31.4 years), with a higher prevalence of females (60%). Sensitization to common allergens was present in 100% of the patients. Rhinitis and asthma were also prevalent comorbidities (80% and 60% of the patients, respectively). Profilin sensitization was not found in any patients and 2 patients were sensitized to PR-10 family protein (40%). Three patients received grass pollen immunotherapy and none of them to birch pollen.

## 5. Discussion

The present study focuses on the development of allergy to new plant foods among nsLTP-sensitized patients. We consider this unresolved issue to be a key point in the management of nsLTP-allergic patients. Our results reveal that 31% of nsLTP-allergic patients became allergic to new plant foods that had been tolerated at the time of diagnosis. We also found that, after 10 years, 13% of patients simply sensitized to nsLTP developed plant-food allergy.

This is a real-life study based on clinical allergy practice. Ten years following diagnosis, a telephone interview was conducted to determine whether patients developed allergies to new plant foods. Real-life studies and the results of the telephone interviews have both advantages and disadvantages which should be considered when interpreting these results. However, we found the results to be valuable as they provide interesting information on the development of allergy to new plant foods, both among patients with nsLTP allergy and among nsLTP-sensitized subjects who have never been allergic to plant foods (latent atopy).

To our knowledge, the report by Asero et al. [19] is the only study designed to evaluate the development of new food allergies in the follow-up of patients allergic to nsLTP. The results of our study, in which 31% of patients developed new plant-food allergies, are in agreement with those of Asero et al. (27%; 18/67 patients), which reinforces the results of both.

A literature search revealed no previous studies addressing the development of plant-food allergy among nsLTP-sensitized patients without previous plant-food allergy. We found that allergy to new plant foods among patients without previous plant-food allergy was not only less frequent, but also less severe, as no patients in the sensitized group had anaphylactic reactions. These data support current recommendations indicating that patients who experienced systemic reactions should always carry auto-injectable adrenaline with them.

Another finding of our study, and one that is found throughout the literature on nsLTP allergy, is that rosaceous fruits and nuts are the foods most frequently responsible for nsLTP allergic reactions [14,19,20], even when discussing new plant-food allergies in the evolution of these patients. We consider this issue relevant, as clinicians should not restrict all nsLTP-allergenic foods in the same way, but rather prioritize the most frequently involved foods when an avoidance diet is necessary.

We also found that sensitization to profilin and PR-10 allergens appears to decrease the risk of severe reactions [21,22], and that nsLTP-specific IgE levels do not predict the occurrence of new plant-food allergy [23], which is consistent with data published in multiple studies.

Asero et al. [19] provided useful recommendations for the management of patients with nsLTP-related food allergy, which we support fully. In addition to these recommendations, we believe nsLTP-allergic patients should undergo risk stratification, as this would allow for tailored management of the heterogeneous and highly variable population of patients with this type of allergy. Specifically, our findings lead us to recommend the following:-Patients should avoid plant foods that provoke allergic reactions after an allergy study based on anamnesis, skin testing, and/or determination of specific IgE and challenge tests when necessary;-Patients with systemic reactions should always carry self-injectable adrenaline on their person;-Additional dietary restrictions should be based on patient risk stratification, as it is impossible to predict severity and/or allergy to new plant foods. In our opinion, key points to stratify the risk of the nsLTP-allergic patients are those appearing in Table 5. Thus, for patients who have developed a systemic reaction to peach peel but who tolerate other foods (even peach pulp), it would be sufficient to avoid peach peel and take self-injectable adrenaline. However, when traveling to the mountains, the countryside, or other remote locales, they should strictly avoid foods related to the nsLTP allergy and be vigilant with NSAIDs and other cofactors, since accessibility to emergency services may be limited and their quality of life would not be significantly altered by such a one-off situation. This is an example of how allergy-management recommendations should be adapted depending on risk stratification.

## 6. Conclusions

In summary, one-third of nsLTP-allergic patients developed allergy to novel plant foods, while one-tenth of nsLTP-sensitized patients without food allergy eventually developed reactions to novel plant foods, which were milder. Finally, risk stratification should be a cornerstone of individualized management for highly varied patients with nsLTP allergy.

## Figures and Tables

**Figure 1 nutrients-13-02165-f001:**
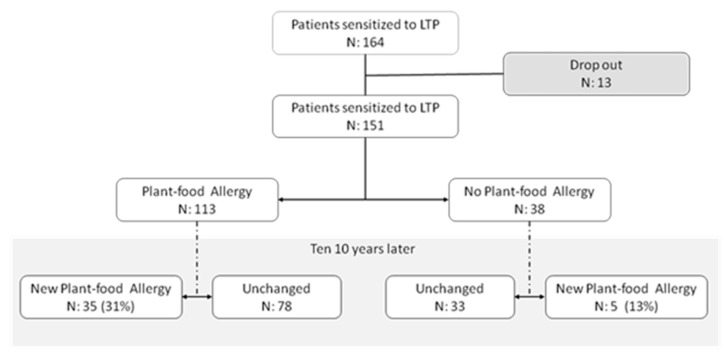
Study flow chart.

**Figure 2 nutrients-13-02165-f002:**
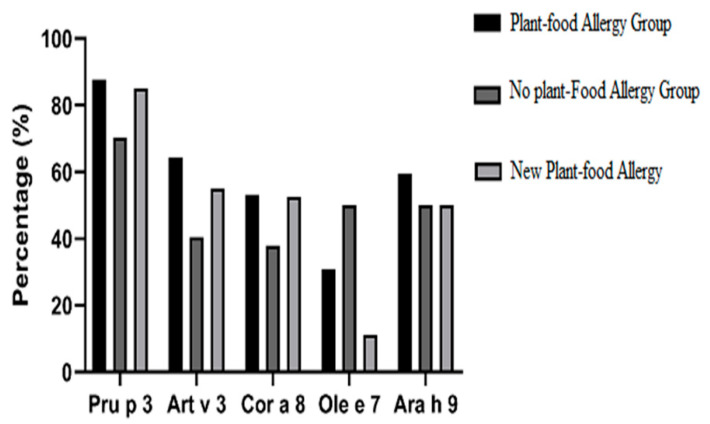
Percentage of sensitization to several nsLTPs in different groups of patients.

**Table 1 nutrients-13-02165-t001:** Demographic and clinical characteristics of patients sensitized to nsLTP (baseline data).

	Food Allergy	Non-Food-Allergy
	Group (*n*:113)	Group (*n*:38)
Sex, male	59 (52.21%)	23 (60.52%)
Age (years) (mean, SD)	31.67 (14.36)	30.87 (12.5)
Previous atopy1 history	94 (83.18%)	28 (73.68%)
Allergic rhinitis	92 (81.41%)	29 (76.32%)
Asthma	58 (51.32%)	18 (47.37%)
Sensitization to common allergens	102 (90.26%)	35 (92.10%)
Pollen sensitization	101 (89.38%)	34 (89.47%)
Grass	92 (81.41%)	29 (76.31%)
Olive	64 (56.64%)	20 (52.63%)
Cypress	58 (51.33%)	18 (47.36%)
Platanus tree	65 (57.52%)	19 (50.00%)
Mugwort	72 (63.72%)	15 (39.47%)
Animal sensitization	47 (41.59%)	22 (57.89%)
Dust mite sensitization	28 (24.77%)	11 (28.95%)
Mold sensitization	21 (18.58%)	8 (21.05%)
Grass pollen immunotherapy	58 (51.33%)	24 (63.16%)
Panallergen sensitization	71 (62.83%)	22 (57.89%)
Profilin	38 (33.63%)	9 (23.68%)
Bet v 1	19 (16.81%)	10 (26.32%)

**Table 2 nutrients-13-02165-t002:** Plant foods involved in the allergic reactions of nsLTP allergy group (*n* = 113) at baseline. Results are shown in number of patients.

Plant Food	Food Allergy	Oral Tolerance	Not Known
*Nuts*	77	30	6
Walnut	54	21	38
Hazelnut	36	41	36
Peanut	36	41	36
Almond	29	48	36
Sunflower seed	15	55	43
***Fruits***	**95**	**18**	**0**
***Rosaceae* fruits**			
Peach	70	37	6
Peach (peel only)	35	37	41
Apricot	22	42	49
Cherry	18	45	50
Strawberry	8	58	47
Plum	20	46	47
***Pomoideae* fruits**			
Apple	37	50	26
Apple (peel only)	54	50	9
Pear	15	90	8
**Other fruits**			
Kiwi	18	67	28
Banana	11	67	35
**Legumes**	**12**	**78**	**23**
Lentil	7	95	11
Bean	4	80	29
Soybean	2	109	2
Chickpea	1	100	12
**Vegetables**	**23**	**89**	**1**
Tomato	12	99	2
Lettuce and derivates	10	84	19
Corn	3	91	19
Eggplant	2	78	35
Cauliflower	2	65	46
**Seed**	**9**	**64**	**40**
Mustard	8	54	51
Sesame	1	69	43
**Cereal** (Wheat)	**2**	**111**	**0**

**Table 3 nutrients-13-02165-t003:** Characteristics of patients non-sensitized to Pru p 3 (*n* = 25).

	Plant-Food-Allergy Group (*n* = 14)	Non-Plant-Food-Allergy Group (*n* = 11)
Sensitization to nsLTP		
Art v 3	7 (50%)	6 (54.54%)
Ara h 9	1 (7.14%)	0 (0%)
Cor a 8	1 (7.14%)	3 (27.27%)
Pla a 3	2 (14.28%)	0 (0%)
Jug r 3	0 (0%)	1 (9.09%)
Cor a 8 + Art v 3	1 (7.14%)	1 (9.09%)
Jug r 3 + Art v 3	1 (7.14%)	0 (0%)
Ara h 9 + Jug r 3 + Art v 3 + Pla a 3	1 (7.14%)	0 (0%)
**Panallergen sensitization**		
Profilin	6 (42.9%)	1 (9.1%)
PR10	3 (21.4%)	2 (18.2%)
**Allergy to new plant food (clinical progression)**	5 (35.7%)	1 (9.1%)

**Table 4 nutrients-13-02165-t004:** Allergy to new plant foods on follow-up study in patients sensitized to nsLTP.

New Plant Food Eliciting Allergy	Plant-Food-Allergy Group (*n* = 35)	Non-Plant-Food-Allergy Group (*n* = 5)
*Rosacea*/*Prunoideae* fruit	7	2
*Rosacea*/*Pomoideae* fruit	3	0
Nuts	7	0
Vegetables	4	0
Cereals	2	0
Legumes	1	1
Seed	1	0
*Rosaceae*/*Prunoideae* fruit & nuts	5	2
Nuts & vegetables	3	0
Nuts & legumes	1	0
*Rosaceae*/*Prunoideae* fruit & legumes	1	0

**Table 5 nutrients-13-02165-t005:** Risk stratification of nsLTP-allergic patients.

Key Points.
- Severity of previous reactions
- New foods most frequently implicated
- Accessibility of emergency services
- Sensitivity to PR-10 and profilin
- Cofactors
- Quality of life

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
