# Peer review of "LTP Allergy Follow-Up Study: Development of Allergy to New Plant Foods 10 Years Later"

_nutrients, 2021, doi:10.3390/nu13072165_

Round 1

Reviewer 1 Report

Major considerations

1-The patients allergic symptoms due to allergy to plant food are not describe neither at the time of recruitment nor at the time of the phone interview.

2-Table 1

-What are "previous atopy history"? Clarify please.

-It is better to add the values of sIgE to nLTP in the table under "Panallergens"

-If you have done Table 1 it is not necessary to describe the table (a bit confusing) so cancel Line 129-141

3-Table 2:

Are these data from the time of recruitment of after the phone call? Clarify please.

What are the "allergic related symptoms"? Clarify please

4- sIgE to nsLTP: it is not clear how may LTP have been tested at the beginning of the study and at the end.

The study design must be improve. It is not possibile to confirm a new allergy to plant food without a challenge test.

The manuscript is confuse to read and not clear.

Author Response

Major considerations

1-The patients allergic symptoms due to allergy to plant food are not describe neither at the time of recruitment nor at the time of the phone interview.

Following the study by Asero et al, symptoms were categorized into: local symptoms, systemic symptoms and anaphylaxis (two or more organs involved) in order to better manage the information and perform a statistical study. This information appears on page 3, lines  114-116.

2-Table 1

-What are "previous atopy history"? Clarify please.

The expression has been changed to previous atopic diseases (Table 1).

-It is better to add the values of sIgE to nLTP in the table under "Panallergens"

We have tried to add the information in lines 139-145 in Table 1, but we found it difficult to do so.The reason was that profilin and Bet v 1 were determined in all patients, while the number of determinations varied for the different nsLTPs, which was explained in lines 104-107.  In addition to the percentage of positive results, the median and Q1-Q3 values are given.

-If you have done Table 1 it is not necessary to describe the table (a bit confusing) so cancel Line 129-141

Thank you. Following the reviewer's suggestion data repeated in table and text were removed.

3-Table 2:

Are these data from the time of recruitment of after the phone call? Clarify please.

These data correspond to the description of the population studied at the beginning of the study, after clinical workup (2009-2011). Following your recommendation this information has been added to the text (Page 2, line 92).

What are the "allergic related symptoms"? Clarify please

There was an error. It has been corrected and should read FOOD ALLERGY, referring to the diagnosis of the food responsible for the allergy. Please see  Page 5, Table 2.

4- sIgE to nsLTP: it is not clear how may LTP have been tested at the beginning of the study and at the end.

LTP was only tested at the beginning of the study and are described  in Page 4, Lines 140-145. At the end of the study, only a telephone interview was conducted to learn about new reactions to foods that had been tolerated at the time of allergy diagnosis. No nsLTP-specific IgE determinations were performed at the end of the study.

The study design must be improved. It is not possibile to confirm a new allergy to plant food without a challenge test.

We agree with the reviewer's opinion, “It is not possible to confirm a new allergy to plant food without a challenge test”. The new plant food allergies were based on the telephone interview conducted by allergists, evaluating the possible cause-effect relationship and the characteristics of the reaction, assessing whether they were compatible with an allergic reaction. However, this type of studies often overestimates the incidence of food allergy. For this reason, we make it clear in “Discussion section” that, results of our study should be read considering the characteristics of this type of studies (real-life study and telephone survey). They have a high sensitivity and negative predictive value, so we can assure that the frequency of new allergies would be less than 30%, but the the positive predictive value is not ideal. This type of studies is less methodologically rigorous, but they are usually carried out to obtain information when there is a lack of data and when it is difficult to carry out the study following more rigorous and complex canons (double-blind placebo study with 10-year follow-up). This information is clearly indicated in paragraphs 245-249. Nonetheless, the results obtained with this methodology were consistent with those previously reported by Asero et al.

The manuscript is confuse to read and not clear.

We hope that the changes made to the manuscript following the reviewers' contributions and the revised version by a native English speaker specialized in medical journals will help to clarify it and make it more readable.

Reviewer 2 Report

Gomez-Lopez and coworkers report the observations coming from a 10-year follow-up of patients hypersensitive to lipid transfer protein. They find that a significant proportion of patients develop new food allergies over time and that this occurs more frequently among those who were already food-allergic than among those who were sensitized only.

GENERAL COMMENTS

Although the study confirms previous data (Ref 18) the study is interesting as the literature on the prognosis in the long term of LTP sensitized patients is rather limited. The manuscript is lengthy and sloppy in several parts and contains a number of grammar errors. A thorough review by a native English speaker is needed.

SPECIFIC POINTS

Line 45. Among the references consider also Int Arch Allergy Immunol. 2009;150(3):271-7.

Line 101. After my reading it seems that patients were diagnosed from 2009 to 2011. Please clarify.

Line 132: What do you mean by the term “atopic”? Are LTP reactors non-atopic?

Results, parts A) and B): these parts are too lengthy. Please be brief and rely on the table without explaining everything in detail point by point.

Table 4: An important point: How did score the foods causing new allergies in-vitro and in-vivo at baseline? In other words, were patients sensitized to all the new offending foods when they were evaluated the first time 10 years before? Were there cases of de-novo sensitization and allergy? Please clarify and discuss.

The discussion is overlong and should be condensed by at least one third.

MINOR POINTS

Regarding the gramma errors some examples follow:

Line 17: which makes it difficult to manage this type of allergy.

Line 19. Patients showing IgE specific for ns LTP….

Line 29: legumes.

Line 30: This was more frequent…

Lines 42-43: Allergy to nsLTP involves several taxonomically unrelated plant-derived foods and heterogenous sensitization profiles and can be….

Line 47. It…. Who/what is the subject of this sentence?

Etc. throughout the entire manuscript.

Author Response

Gomez-Lopez and coworkers report the observations coming from a 10-year follow-up of patients hypersensitive to lipid transfer protein. They find that a significant proportion of patients develop new food allergies over time and that this occurs more frequently among those who were already food-allergic than among those who were sensitized only.

GENERAL COMMENTS

Although the study confirms previous data (Ref 18) the study is interesting as the literature on the prognosis in the long term of LTP sensitized patients is rather limited. The manuscript is lengthy and sloppy in several parts and contains a number of grammar errors. A thorough review by a native English speaker is needed.

We are grateful for the reviewer's comments. Following the reviewer's recommendations, the manuscript has been revised by a native English speaker specialized in medical journals.

SPECIFIC POINTS

Line 45. Among the references consider also Int Arch Allergy Immunol. 2009;150(3):271-7.

The reference has been added (Please see reference #9).

Line 101. After my reading it seems that patients were diagnosed from 2009 to 2011. Please clarify.

  1. The allergy study and diagnosis were performed between 2009 and 2011. This information appears on Page 2, Line 92.

Line 132: What do you mean by the term “atopic”? Are LTP reactors non-atopic?

You are right. The term “atopic” has been removed and changed by “sensitization to common allergens”

Results, parts A) and B): these parts are too lengthy. Please be brief and rely on the table without explaining everything in detail point by point.

Thank you very much. Following the reviewer's suggestions, both parts (A & B) have been shortened and the repeated information in tables excluded from the text.

Table 4: An important point: How did score the foods causing new allergies in-vitro and in-vivo at baseline? In other words, were patients sensitized to all the new offending foods when they were evaluated the first time 10 years before? Were there cases of de-novo sensitization and allergy? Please clarify and discuss.

No. The initial allergology study was a real-life study, in the clinical practice of the Allergy Service of Fundación Jiménez Díaz. In our usual practice we give little value to the results of skin tests or specific IgE to complete plant food extracts, for two reasons: food extracts are not standardized and cross-reactivity among nsLTPs from different plant foods is extensive. Nonetheless, we perform skin prick by prick tests to the natural food if the allergist considers it necessary.

The discussion is overlong and should be condensed by at least one third.

Thank you very much. Following the reviewer's recommendations, the discussion has been shortened.

MINOR POINTS

Regarding the gramma errors some examples follow:

Line 17: which makes it difficult to manage this type of allergy.

Line 19. Patients showing IgE specific for ns LTP….

Line 29: legumes.

Line 30: This was more frequent…

Lines 42-43: Allergy to nsLTP involves several taxonomically unrelated plant-derived foods and heterogenous sensitization profiles and can be….

Line 47. It…. Who/what is the subject of this sentence?

Etc. throughout the entire manuscript.

Many thanks again. Following the reviewer's suggestions, the grammatical errors have been corrected and the full manuscript has been revised by a native English speaker specialized in medical journals.

Reviewer 3 Report

Dear authors,

This paper describes the development of allergy to new plant foods 10 years after sensitization to LTP. During a follow-up telephone interview the patients were asked for new allergic reactions to other plant foods. The results are interesting and give insight in the process of LTP allergy and development of such allergy over the years. Nevertheless, most conclusions are based on “allergic related symptoms” and ISAC results. This is rather a vague description and lacks e.g. a thorough food specific symptom score, e.g. dose, time, severity etc. In this form the paper is not suitable for Publication and major revisions are necessary. Furthermore, There are numerous spelling errors, abbreviation errors, and inconsequent and vague definitions. Furthermore, the English should be corrected throughout the document. 

Major revisions needed:

  • The results from 10 years earlier were apparently not published and extra information cannot be found in this article. Definition of being allergic to plant foods was based on anamnesis, skin test or specific IgE and challenge test”, but the article lacks these results. So, the two groups: “nsLTP allergy group and the non-food-allergy group” are not well described. “Allergic related symptoms” is absolutely too vague and besides it should be “allergy related symtpoms”. An IgE mediated food allergy is characterized by symptoms of Skin, GI, Lung, OAS, or a combination. The authors report “75.2% systemic reactions”. The kind of reactions should be reported in the article.
  • 81% of the patients are Grass pollen allergic and 20% are Birch pollen allergic, with 80.13% Rhinitis – and 50.33% asthma symptoms. Most likely, these patients (mean age 31) are eligible for Immunotherapy (AIT). Nothing is mentioned about this, which is in fact strange because in some cases AIT might have effect on the food allergy.
  • During the telephone interview ten years later, the most interesting question would be how the earlier reported food allergy has developed over the years. Do they still have a diet on those foods, did they experience more or less allergic reactions, etc. Why didn’t the authors ask this to the patients? All results are now based on allergy to foods 10 years ago, but that might have changed over the years.
  • The follow-up consisted in only a telephone interview. There is no current information on sensitization to the “new plant food allergy”. Only asking the patient for symptoms is quite risky, because from earlier studies we know that patients tend to define their subjective symptoms as an allergy and it is the patient’s perception to be allergic.

Minor revisions needed:

Throughout the manuscript recombinant allergens are spelled wrong or different with and without spaces: Cora8 and Artv3,Arah9, Jugr3, Artv3 or Pru p 3, Cor a 8, Art v 3. Same goes for: LTP or nsLTP or nsLTPS or nsLTPs

#91 needs a space after 38

#96 figure1: Allergy to new foods N: 40 is NOT correct. This should be 35 (31%)

#108 Define “possible allergic condition”

#128 atopy1?? In table 1

#134 space before 81%

#134 sensitized to grass sensitization ???

#150 responsible of??? Anaphylactic reacton??

# 152 consensitization??

#156 allergic related symptoms should be changed throughout the manuscript in: allergy related symptoms AND authors should add the definition!!

#159 and were the group of ???

#160 subject??? Or subjects ???

#168 add data, numbers and p values!!. Only “significant or not significant” is not enough

#203/204 add data , numbers and p values!!. Only “significant or not significant” is not enough

# 220 OVER the follow-up study

#304 add sensitization to the specific food allergen???

Author Response

Dear authors,

This paper describes the development of allergy to new plant foods 10 years after sensitization to LTP. During a follow-up telephone interview the patients were asked for new allergic reactions to other plant foods. The results are interesting and give insight in the process of LTP allergy and development of such allergy over the years. Nevertheless, most conclusions are based on “allergic related symptoms” and ISAC results. This is rather a vague description and lacks e.g. a thorough food specific symptom score, e.g. dose, time, severity etc. In this form the paper is not suitable for Publication and major revisions are necessary. Furthermore, There are numerous spelling errors, abbreviation errors, and inconsequent and vague definitions. Furthermore, the English should be corrected throughout the document. 

Major revisions needed:

  • The results from 10 years earlier were apparently not published and extra information cannot be found in this article. Definition of being allergic to plant foods was based on anamnesis, skin test or specific IgE and challenge test”, but the article lacks these results. So, the two groups: “nsLTP allergy group and the non-food-allergy group” are not well described. “Allergic related symptoms” is absolutely too vague and besides it should be “allergy related symtpoms”. An IgE mediated food allergy is characterized by symptoms of Skin, GI, Lung, OAS, or a combination. The authors report “75.2% systemic reactions”. The kind of reactions should be reported in the article.

Thank you. There was a mistake in the table and where it says: "Allergic related symptoms" it should read "Food Allergy" and refers to the diagnosis after performing the allergy study in the Allergy Service of the Fundación Jiménez Díaz in real life conditions. The table refers to the food causing allergy (Food Allergy).

When the data were collected and similar to the manuscript by Asero et al, symptoms were grouped into local symptoms and systemic or general symptoms. If several organs were affected, anaphylaxis was considered.

In this study there was a high percentage of systemic reactions, which is due to selection bias. The selection criterion was to have a positive nsLTP result by ISAC, which is clearly indicated in material and method.  We request ISAC instead of immunoCAP in complex patients.

  • 81% of the patients are Grass pollen allergic and 20% are Birch pollen allergic, with 80.13% Rhinitis – and 50.33% asthma symptoms. Most likely, these patients (mean age 31) are eligible for Immunotherapy (AIT). Nothing is mentioned about this, which is in fact strange because in some cases AIT might have effect on the food allergy.

Thank you again for your comments. This information has been added to the text. Please see Page 4, Table 1; Page 8, lines 221-222 and Page 8, lines 236-237.

No patients in the study received immunotherapy to birch pollen extract, becasue Madrid is an area with low or absent birch pollen counts.

  • During the telephone interview ten years later, the most interesting question would be how the earlier reported food allergy has developed over the years. Do they still have a diet on those foods, did they experience more or less allergic reactions, etc. Why didn’t the authors ask this to the patients? All results are now based on allergy to foods 10 years ago, but that might have changed over the years.

The evolution of LTP allergy has two very interesting issues: a) to know if LTP allergy disappears with time and how long after, and b) to know the development of allergy to new plant foods. 

The study focused on finding out allergy to new foods. What the reviewer proposes is a very interesting study to know whether LTP allergy is resolved over time and after how long. We have not approached it because we consider that it requires a different design and should include a more in-depth evaluation, in the allergist's office (not by telephone interview) and with oral food challenge tests, which is beyond the scope of this study design.

Even so, we consider that the data provided in the study are very interesting for the management of patients allergic to LTP.

  • The follow-up consisted in only a telephone interview. There is no current information on sensitization to the “new plant food allergy”. Only asking the patient for symptoms is quite risky, because from earlier studies we know that patients tend to define their subjective symptoms as an allergy and it is the patient’s perception to be allergic.

We agree with the reviewer's opinion, “It is not possible to confirm a new allergy to plant food without a challenge test”. The new plant food allergies were based on the telephone interview conducted by allergists, evaluating the possible cause-effect relationship and the characteristics of the reaction, assessing whether they were compatible with an allergic reaction. However, this type of studies often overestimates the incidence of food allergy. For this reason, we make it clear in “Discussion section” that, results of our study should be read considering the characteristics of this type of studies (real-life study and telephone survey). They have a high sensitivity and negative predictive value, so we can assure that the frequency of new allergies would be less than 30%, but the positive predictive value is not ideal. This type of studies is less methodologically rigorous, but they are usually carried out to obtain information when there is a lack of data and when it is difficult to carry out the study following more rigorous and complex canons (double-blind placebo study with 10-year follow-up). This information is clearly indicated in paragraphs 245-249. Nonetheless, the results obtained with this methodology were consistent with those previously reported by Asero et al.

Minor revisions needed:

Throughout the manuscript recombinant allergens are spelled wrong or different with and without spaces: Cora8 and Artv3,Arah9, Jugr3, Artv3 or Pru p 3, Cor a 8, Art v 3. Same goes for: LTP or nsLTP or nsLTPS or nsLTPs

#91 needs a space after 38

#96 figure1: Allergy to new foods N: 40 is NOT correct. This should be 35 (31%).

Thank you. There was a mistake and it has been changed

#108 Define “possible allergic condition”

#128 atopy1?? In table 1

#134 space before 81%

#134 sensitized to grass sensitization ???

#150 responsible of??? Anaphylactic reacton??

# 152 consensitization??

#156 allergic related symptoms should be changed throughout the manuscript in: allergy related symptoms AND authors should add the definition!!

#159 and were the group of ???

#160 subject??? Or subjects ???

#168 add data, numbers and p values!!. Only “significant or not significant” is not enough

#203/204 add data , numbers and p values!!. Only “significant or not significant” is not enough

# 220 OVER the follow-up study

Thank you again. Following your suggestions changes have been done.

#304 add sensitization to the specific food allergen???

We are sorry, but unfortunately, we do not understand the recommendation.

Round 2

Reviewer 1 Report

Accept in present form
